# The effect of human mobility restrictions on the COVID-19 transmission network in China

**Tatsushi Oka**[ID]◐*, **Wei Wei**◐, **Dan Zhu**◐

Department of Econometrics and Business Statistics, Monash University, Caulfield, Victoria, Australia

◐ These authors contributed equally to this work.
* tatsushi.oka@monash.edu

## Abstract

### Background

COVID-19 poses a severe threat worldwide. This study analyzes its propagation and evaluates statistically the effect of mobility restriction policies on the spread of the disease.

### Methods

We apply a variation of the stochastic Susceptible-Infectious-Recovered model to describe the temporal-spatial evolution of the disease across 33 provincial regions in China, where the disease was first identified. We employ Bayesian Markov Chain Monte-Carlo methods to estimate the model and to characterize a dynamic transmission network, which enables us to evaluate the effectiveness of various local and national policies.

### Results

The spread of the disease in China was predominantly driven by community transmission within regions, which dropped substantially after local governments imposed various lockdown policies. Further, Hubei was only the epicenter of the early epidemic stage. Secondary epicenters, such as Beijing and Guangdong, had already become established by late January 2020. The transmission from these epicenters substantially declined following the introduction of mobility restrictions across regions.

### Conclusions

The spatial transmission network is able to differentiate the effect of the local lockdown policies and the cross-region mobility restrictions. We conclude that both are important policy tools for curbing the disease transmission. The coordination between central and local governments is important in suppressing the spread of infectious diseases.

**Data Availability Statement:** All relevant data are within the manuscript and its Supporting information files.

**Funding:** We received financial support from the Centre for Development Economics and

Sustainability (CDES) to hire a research assistant for collecting and cleaning data. In that sense, our research is partially supported by the CDES and the remaining parts are conducted by ourselves without any financial supports.

**Competing interests:** No authors have competing interests.

## Introduction

The ongoing pandemic of coronavirus disease of 2019 (COVID-19) poses a threat to public health and has disrupted economic activities globally. Although there are limited policy tools available to stem the disease spread, restricting human mobility though lockdown or border closure policies was identified as an effective measure. Simply put, the virus itself cannot move anywhere without assistance. In many countries, mobility restriction led to the containment of the virus's spread. Given the importance of mobility restriction as an effective policy, it is critical to quantify its effects.

In this study, we consider a multivariate discrete-time Markov model to analyze the propagation of COVID-19 across 33 provincial regions of China. Thereby, we allow for heterogeneous disease transmission both within and across regions. Our dataset includes 27 provinces, four municipalities (Beijing, Shanghai, Tianjin, and Chongqing), and two special administrative regions (Hong Kong and Macao). Through the paper, we use "region" for the provinces, municipalities and special administrative regions. Our model takes into account human mobility as a key driver of disease transmission across regions and identifies epicenters of disease propagation, as well as the effect of mobility restrictions on infection rates. We extract information on daily human mobility across regions from January 11 to March 15, 2020, from the Baidu database [1] and apply the Bayesian framework to estimate the model. The sampling period in use for our analysis exhibits substantial exogenous variations in human mobility rates due to the high number of movements around Chinese New Year (January 25) and a sudden decline in movements after policy interventions were introduced. We evaluate the effect of mobility restrictions on the disease spread between regions by comparing outcomes under actual and counterfactual human mobility, which is extracted from the 2019 data.

Our empirical results document substantial heterogeneity in the rate of infection across regions. The results also demonstrate the effectiveness of the lockdown policy in curbing the spread of the pandemic. The transmission mechanism of the disease in China is found to be predominately community transmission within all regions. Further, our analysis based on the 2019 mobility data suggests that the external transmission would not have been suppressed if people had continued to be allowed to move freely across regional borders as usual. Interestingly, our results show that Hubei is not the only epicenter of the early epidemic stage. Other epicenters, such as Beijing and Guangdong, had already become established by late January 2020. The pandemic radiated out to the subordinate regions of these cities with varying degrees of severity. Our approach sheds light on the evolution of the transmission network over time and provides useful insight into the formulation of lockdown policies amid the pandemic.

The methodological part of the paper draws on and contributes to several literatures. First, since the outbreak of COVID-19, many studies have provided simulations and predictions using a deterministic susceptible-infective-recovered (SIR) model in [2]. The SIR model divides a well-defined population into three compartments, namely susceptible, infective, and recovered individuals, and characterizes disease transmission as individuals' transition between these compartments [3]. As [4] discusses, however, stochastic modeling of epidemics is essential when the number of infectious individuals is small, and the transition between the compartments depends on demography and the environment. We consider a variation of a stochastic SIR model for the 33 regions in China.

Second, the most critical feature of our model is that it captures the impact of human movements on spatio-temporal disease transmission. The quantitative modeling of human movements has a long-standing history in fields like transportation, tourism, and urban planning.

The use of the gravity model has been popular in these fields [5–7] as well as in the field of economics [8, 9]. In the epidemiology literature, the gravity model was first applied by [10]. Gravity-type models are also widely adopted in more recent studies [11–14]. Alternatively, the radiation model, proposed in [15], is used to predict spatial disease transmission [16, 17]. Both the gravity and radiation models treat the transition probabilities of individuals from one place to another as a function of population sizes and geometric distances, both of which are almost invariant on a daily basis. By contrast, this study uses known information on daily human mobility to characterize disease transmission across regions and evaluate the dynamic impact of mobility restrictions.

Third, there is a growing body of literature dedicated to the study of the spread of infections in China. The transmission of COVID-19 from Wuhan to other cities is studied in [18]. They combine three data sets: 1) the monthly number of domestic and international flight bookings from Wuhan in January to February 2019, 2) the number of daily domestic passengers by train and car, and 3) travel volumes forecast from and to Wuhan by Wuhan Municipal Transportation Management Bureau [19]. Use human mobility information from Baidu-Qianxi and analyze the disease spread from Wuhan to other regions between January 1 and February 10, 2020. They predict daily case counts in the early phase of disease spread using three different models: Poisson, negative binomial, and log-linear regression. Both [18] and [19] document the significance of human mobility from Wuhan in causing the spread of the disease in the early phase. Both authors also underscore that the effect travel restrictions in Hubei had on containing the spread of the disease. In our study, we estimate a model that accounts for disease transmission across all regions, using data spanning from the beginning of the epidemic until the end of the first wave in China. Our research complements the existing research by providing a more complete understanding of the spread of the disease using a broad and well-defined framework.

Lastly, we contribute to the large body of literature that analyzes infection control in epidemiology [20, 21] and in economics [22–25]. The economic literature theoretically analyzes the optimal control of infection from the perspective of a social planner and discusses how public policies, such as subsidies or taxes, can provide individuals with the required incentives to achieve the social planner's first-best solution. We analyze the within-region policies and the cross-region mobility restrictions separately and provide insight to the importance of coordination between local and central governments.

Several conclusions can be drawn from our empirical results. First, local lockdown policies are very effective in suppressing the spread of infectious diseases. Second, disease transmission also responds well to mobility restrictions. Third, our spatial transmission network reveals that regional epicenters can quickly become established and transmit the disease to connected regions. Thus, local government interventions, such as lockdown in Wuhan, cannot fully contain the disease. If the primary goal is to eliminate the disease entirely, the central and local governments must implement preventive measures simultaneously. Furthermore, the coordination between the local and national governments can facilitate the smooth enforcement of COVID-related policies and the promotion of hygiene practices by the general public.

## Model

This section first introduces the variation of the susceptible-infective-recovered model applied in this study. Subsequently, it explains the specification of internal and external disease transmission in the model.

## Stochastic SIR model with spatial effects

We apply a variation of stochastic SIR model to describe the evolution of three variables: $S_{jt}$, $I_{jt}$ and $R_{jt}$, which denote the number of susceptible, infective, and recovered individuals in region $j$ at time $t$, respectively. Also, let $D_{jt}$ denote the cumulative number of deaths by $t$ and let $N_j$ be the total population in region $j$. Then, we have the following identity: $N_j = S_{jt} + I_{jt} + R_{jt} + D_{jt}$. Our formulation has kept the regional population sizes time-invariant. As a robustness check, we estimate our model, allowing for time-variant regional population sizes due to travelers across regions. Under the different formulation, we obtain similar empirical results, which are available upon request. We observe regional panel data of $(I_{jt}, R_{jt}, D_{jt}, N_j)$ for region $j = 1, \ldots, J$ and time $t = 0, \ldots, T$ with $J$ and $T$ denoting the sample size of regions and time periods, respectively. In what follows, we use $\mathcal{F}_t$ to denote the available information set at time $t$.

We denote by $\Delta^I_{j,t+1}$ the number of transitions from susceptible to infected states in region $j$ at time $t + 1$. The number of newly infected individuals $\Delta^I_{j,t+1}$ is assumed to be a random variable following the Poisson distribution conditional on $\mathcal{F}_t$, with the conditional mean given by

$$\mathbb{E}[\Delta^I_{j,t+1}|\mathcal{F}_t] = \left(\beta_{jt}\frac{I_{jt}}{N_j} + \lambda_{jt}\right)S_{jt}. \tag{1}$$

At its core, the equation above follows the Bass model [26], which was originally proposed for describing the diffusion of new products. The key feature of the Bass model is that the acceptance of a new product is driven by either internal influences, such as contagious adopters to which other individuals are connected, or external influences, such as mass media or commercials. The distinction between internal and external influences is adopted by [27] in a deterministic SIR model. Similarly, we can interpret the term $\beta_{jt} I_{jt}/N$ as region $j$'s internal infection rate, which depends on the proportion of infected individuals $I_{jt}/N$ and the internal transmission rate $\beta_{jt}$. Further, we consider the term $\lambda_{jt}$ as the external infection rate, which reflects the rate of infection attributable to transmission from outside of region $j$. If the border to region $j$ is closed, the external effect $\lambda_{jt}$ equals zero, and the model becomes the standard stochastic SIR model [28].

To describe the state transition from the infected state, we use a Markov chain model in which infected individuals either remain infected or move to another state: recovery or death. More specifically, let $\Delta^R_{j,t+1} := R_{j,t+1} - R_{jt}$ and $\Delta^D_{j,t+1} := D_{j,t+1} - D_{jt}$ be changes in the number of recoveries and deaths, respectively. We assume that the transition probability from the infected state at time $t$ follows a multinomial distribution conditional on $\mathcal{F}_t$, satisfying that $\mathbb{E}[\Delta^R_{j,t+1}|\mathcal{F}_t] = \gamma I_{j,t}$ and $\mathbb{E}[\Delta^D_{j,t+1}|\mathcal{F}_t] = \delta I_{j,t}$. Here, the parameters $\gamma$ and $\delta$ are used to represent the recovery and death rates, respectively. As a robustness check, we have considered heterogeneous parameters that differentiate the first epicenter, Hubei, from the rest of China. The empirical result suggests a qualitatively similar conclusion.

Given the stochastic transition among all states, the number of infected and susceptible individuals at time $t + 1$ are given by the following state equations:

$$I_{j,t+1} = I_{jt} + \Delta^I_{j,t+1} - \Delta^R_{j,t+1} - \Delta^D_{j,t+1}, \tag{2}$$

$$S_{j,t+1} = S_{jt} - \Delta^I_{j,t+1}. \tag{3}$$

## Internal transmission

The internal transmission rate $\beta_{jt}$ measures to what extent contacts between an infected individual and the susceptible population at time $t$ leads to the transmission of the pathogen. Thus, it can be interpreted as the number of "effective" contacts. We allow for $\beta_{jt}$ to vary per region and across time. This is because the contact frequency depends on region-specific characteristics, such as population density, as well as time-varying factors, such as policy intervention (e.g. contact tracing and forced quarantine) and behavior changes (e.g. better hygiene practices and social distancing). In China, almost all local governments declared the top-level state of emergency in the early phase of the pandemic (January 23–25, 2020), which effectively induced changes in individuals' behavior. Thus, we assume that intervention by local governments affects internal transmission gradually. Specifically, we consider the following specification:

$$\log \beta_{jt} = \log \beta_{j,t-1} + \alpha_j X_{j,t-h}, \tag{4}$$

where $X_{j,t-h}$ is an observed dummy variable taking the value of 1 if the local government in region $j$ has activated the top-level health emergency response at time $t - h$ and 0 otherwise. This $X_{j,t-h}$ reflects the implementation of various intervention policies that we collectively call the *lockdown policy*. We consider a lag $h > 0$ to account for lagged effects of the policy intervention and set four days ($h = 4$) for our estimation. In the existing literature the mean incubation period of COVID-19 is estimated as roughly 5 days, see [19, 29] among others.

The parameter $\alpha_j$ is allowed to be heterogeneous across regions, reflecting different measures taken by local governments and regional characteristics. The time-varying parameter $\beta_{jt}$ in (4) depends on the initial value $\beta_{j,0}$ and the response to the intervention $\alpha_j$. Our specification allows $\beta_{jt}$ to approach zero in consideration of the draconian measures adopted in China and the suppression of the disease in the first wave. Alternatively, $\beta_{jt}$ could be set to approach a non-zero value as in [30]. Their dynamics can be considered as a special case of the transfer function model in [31] for intervention analysis. See [32, 33] among others for analysis of other type policies.

We specify a hierarchical structure for the transmission parameters across regions, by using a bivariate normal distribution: $(\log \beta_{j,0}, \alpha_j)' \sim N(\mu, \Sigma)$ with mean $\mu := (\mu_\beta, \mu_\alpha)'$ and variance matrix $\Sigma$. Under this specification, the average of the internal transmission rate without any control is given by $\mathbb{E}[\beta_{j,0}] = \exp(\mu_\beta + 1/2\Sigma_{11})$ with $\Sigma_{11}$ denoting the (1,1)-element of $\Sigma$, while the effect of intervention on average is given by $\mathbb{E}[\alpha_j] = \mu_\alpha$.

## External transmission

Using Baidu's daily mobility data [1], we construct a measure of the "intensity" of the disease transmission between regions. The mobility data includes an outflux mobility index for all regions and details the proportion of travelers between regions. We use $M_{kt}^{out}$ to denote the outflux mobility index in region $k$ at time $t$ and we use $P_{kjt}$ to represent the proportion of travelers from region $k$ to region $j$ at time $t$. The mobility index $M_{kt}^{out}$ represents a relative strength measure of the outflux, which is scaled by Baidu's proprietary method, rather than the numbers of outflux. This index is comparable across regions and time. The change of $M_{kt}^{out}$ from its standard level reflects *mobility restrictions*. Additionally, we observe the proportion of daily travelers $P_{kjt}$ between the 31 mainland regions in the sample of 33 provincial regions, which means Hong Kong and Macao are excluded. We impute the entries for Hong Kong and Macao based on the radiation model [15]. We find that the prediction of influx based on the imputed $P_{kjt}$ value traces the index of human influx well and also outperforms the prediction using only the radiation model.

We use $M_{kt}^{out} P_{kjt}$ to measure the (scaled) flux from origin $k$ to destination $j$ and then construct an "intensity" of infected flux from origin $k$ to destination $j$ at time $t$ by $M_{k,t-h}^{out} P_{kj,t-h}(I_{kt}/N_k)$ with a lag $h > 0$ in the mobility measure. As there is a time lag between getting infected and showing symptoms, our formulation takes into account that travelers from origin $k$ at time $t - h$ face case counts $I_{kt}$, which are recorded at $t$. Given the "intensity" of daily infected flux, we consider the external infection rate in region $j$ at time $t$ as follows:

$$\lambda_{jt} = \frac{\theta_{jt}}{N_j} \sum_{k \neq j} M_{k,t-h}^{out} P_{kj,t-h} \frac{I_{kt}}{N_k}, \tag{5}$$

The time-varying parameter $\theta_{jt}$ reflects the strength of external transmission to region $j$ and also normalizes the unit because the index $M_{kt}^{out}$ is a scaled measure. As in the specification for $\beta_{jt}$, we allow $\theta_t$ to respond to policy intervention gradually, i.e., $\log \theta_{jt} = \log \theta_{j,t-1} + \rho X_{j,t-h}$, where $\rho$ is a parameter.

## Data and estimation method

### Data

We use the daily data on COVID-19 infection and individuals' mobility from January 11 to March 15, 2020. The daily data of the infection, death, and recovery cases for each region are obtained from the National Health Commission of China and its affiliates (http://www.nhc.gov.cn/xcs/yqtb/list_gzbd.shtml). The human mobility data is obtained from Baidu Migration [1]. Both datasets are publicly accessible. The data provides a daily outflux index for each of the 33 regions as well as the destinations of the outflux. For our counterfactual analysis, we use the mobility data set of 2019 from Baidu-Qianxi matched according to the Chinese New Year. The plots of the outflux in both 2020 and 2019 are shown in Fig 1. The outflux indices before the Chinese New Year in both 2019 and 2020 are dominated by regions such as Guangdong, Zhejiang, and Beijing. It is expected that most workers would be leaving these areas to return for their home regions for the holiday. For Hubei, the outflux was moderate in both years. The outflux reduced to a negligible level at the time when the lockdown policy prevailed.

### Estimation method

We adopt a Bayesian Markov Chain Monte-Carlo framework for estimation. Given the information on infection, recovery and death cases, we can estimate our model separately for the infection and the recovery and death. In our model, the number of recovered and death cases follows a multinomial distribution. Thus, the likelihood of the parameters of recovery rate $\gamma$ and death rate $\delta$ has an analytic form. We use a standard random-walk Metropolis sampler with uninformative prior.

For the new case counts following the Poisson distribution, we simulate the posterior distribution using the algorithm in [34], which is based on data augmentation and a Metropolis-Hastings-within-Gibbs sampler. We divide the set of parameters into $J + 2$ blocks: $\{(\log \beta_{j,0}, \alpha_j)\}_{j=1}^{J}$, $(\mu, \Sigma)$, and $(\log \theta_0, \rho)$, and then sample sequentially using their conditional posteriors. For each block of $\{(\log \beta_{j,0}, \alpha_j)\}_{j=1}^{J}$, we use a multivariate-$t$ proposal density whose mean and covariance are computed from the mode and Hessian of the conditional posterior. For $(\mu, \Sigma)$, we specify a Gaussian-inverse Wishart prior, $NIW(\mu^*, \kappa^*, \Lambda^*, \nu^*)$, with $\mu^* = (-1, -0.1)'$, $\kappa^* = 1$, $\Lambda^* = \text{diag}(1, 0.05)$, and $\nu^* = 10$. This prior is weakly informative in $\mu$ and moderately informative in the variance matrix $\Sigma$. Lastly, the block $(\log \theta_0, \rho)$ is updated using a

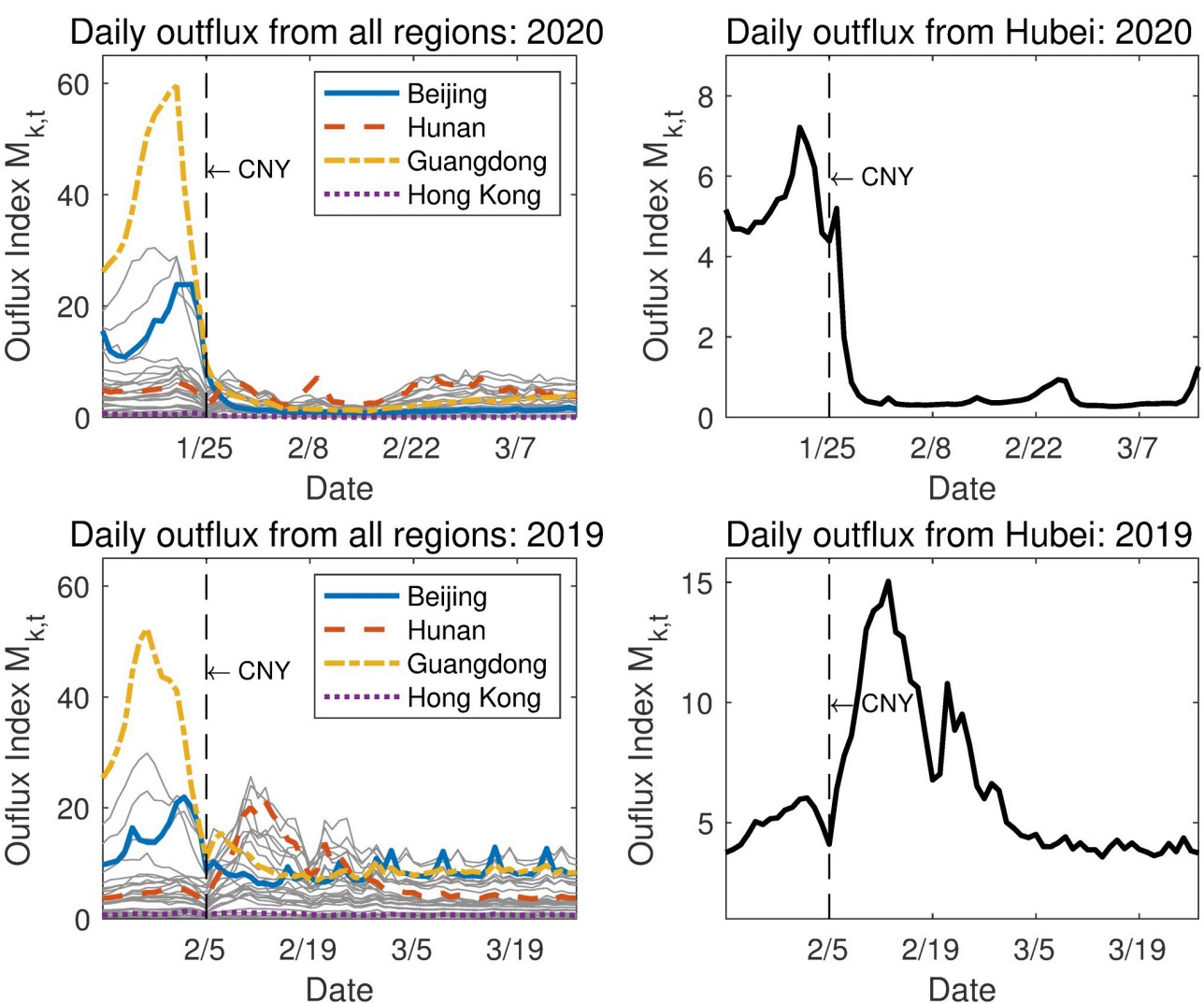

**Fig 1. Daily outflux in 2020 and 2019.** The two panels on the left column show daily outflux from all regions in 2020 and 2019. The ones on the right column show the outflux only in Hubei. In each panel, a dashed vertical line shows the date of the Chinese New Year (CNY) in 2019 and 2020.

Gaussian prior $N(\pi, \Omega)$ with $\pi = (0.1, -0.1)'$ and $\Omega = \text{diag}(0.1, 0.1)$. As the posterior of $(\log \theta_0, \rho)$ depends on all $J \times T$ observations, the contribution of the prior is minimal.

### China's practices to contain COVID-19 transmission

China was the first nation to face the health challenge presented by COVID-19. The situation was exacerbated as the outbreak coincided with the Chinese New Year (CNY), being the single event with the world's most significant population movement. China took many measures to contain its transmission, including extending the CNY holidays, rapid case diagnosis, strict quarantine of close contacts with follow-up checks, and active case surveillance such as requiring daily health declaration from essential workers (see [35] for a systematic review of China's COVID-19 measures.)

Most regions in China adopted similar measures, which we collectively call lockdown policies, to curb the intra-region transmission. The exact implementation and the timing varies between regions. The strictest policies were implemented in Hubei. Wuhan City was

completely locked down on January 23, and was soon followed by the other cities in Hubei Province. Each household was only allowed one person to shop for necessities every two or three days in these cities. In some communities (e.g. Fuxing-cheng Community in Xianning City, Hubei Province), no one was permitted to go out, and daily necessities were instead delivered by government-assigned personnel.

Hong Kong and Macao did not impose the same harsh measures as mainland China. For example, Hong Kong only imposed suspension of schools, social distancing and wearing facemasks in public [36, 37]. Hong Kong also closed only three border control points on 5 February, with the Hong Kong-Zhuhai-Macao Bridge Control Point remaining open.

Measures for reducing inter-regional transmission were imposed at the central government level. The Civil Aviation Administration of China cut the number of flights to Hubei Province since its official lockdown date. For other regions of China, halting long-distance buses/trains and reducing the frequencies of bus/train services were implemented to curb the population movement.

Both local and central government bodies put a large amount effort to health promotion and education. The public paid unprecedented attention to self-protection. Hence, even for regions where the government did not strictly impose facemasks, people would wear a facemask and use sanitisers in public. The decrease in cross-region mobility is largely due to travelers acting precautionarily and voluntarily canceling their travel plans.

## Empirical results

This section first presents the estimation result for the heterogeneous internal infection rate and the effect of the regional intervention. We then compare the results of internal and external infection and provide additional findings based on a transmission network between regions.

### Transmission parameter across regions

In Fig 2, we present the estimation result of internal transmission rates. Panel (a) of Fig 2 shows the posterior means of the initial transmission parameter, $\beta_{j,0}$. The significant heterogeneity in the initial infection rate is evident here. Hubei has the highest value with a very tight posterior credible interval. Panel (b) of Fig 2 reports the transition of posterior means of the internal transmission rate $\beta_{j,t}$, which depicts the effects of policy intervention. The top-level health emergency response was activated for January 23–25, 2020, in all regions, except Xizang, which went into the state of emergency on January 30, 2020. As in Eq (4), the number of new infections is shown to be affected by the policy implemented five days before. In Fig 2, the effect of the intervention is evident but not immediate; it shows that for most regions, it took 4 to 7 days for $\beta_{jt}$ to decrease to half of its original value. The posterior mean of recovery rate $\gamma$ is 4.15% with a 95% credible interval (4.11%,4.18%). The posterior mean of death rate $\delta$ and 0.213%, with a 95% credible interval (0.206%,0.220%).

### Transmission rates: External versus internal

In combination with the lockdown policy, it is important to specifically study human mobility in the context of the COVID-19 pandemic. Our analysis decomposes the expected number of infections into infections resulting from internal and external transmission for all regions. Fig 3 presents results for four regions, each of which represents a different region, but all have similar characteristics. Namely, we consider megacities (Beijing), the neighboring regions of Hubei (Hunan), the secondary epicenters (Guangdong), and the special administrative regions outside mainland China (Hong Kong).

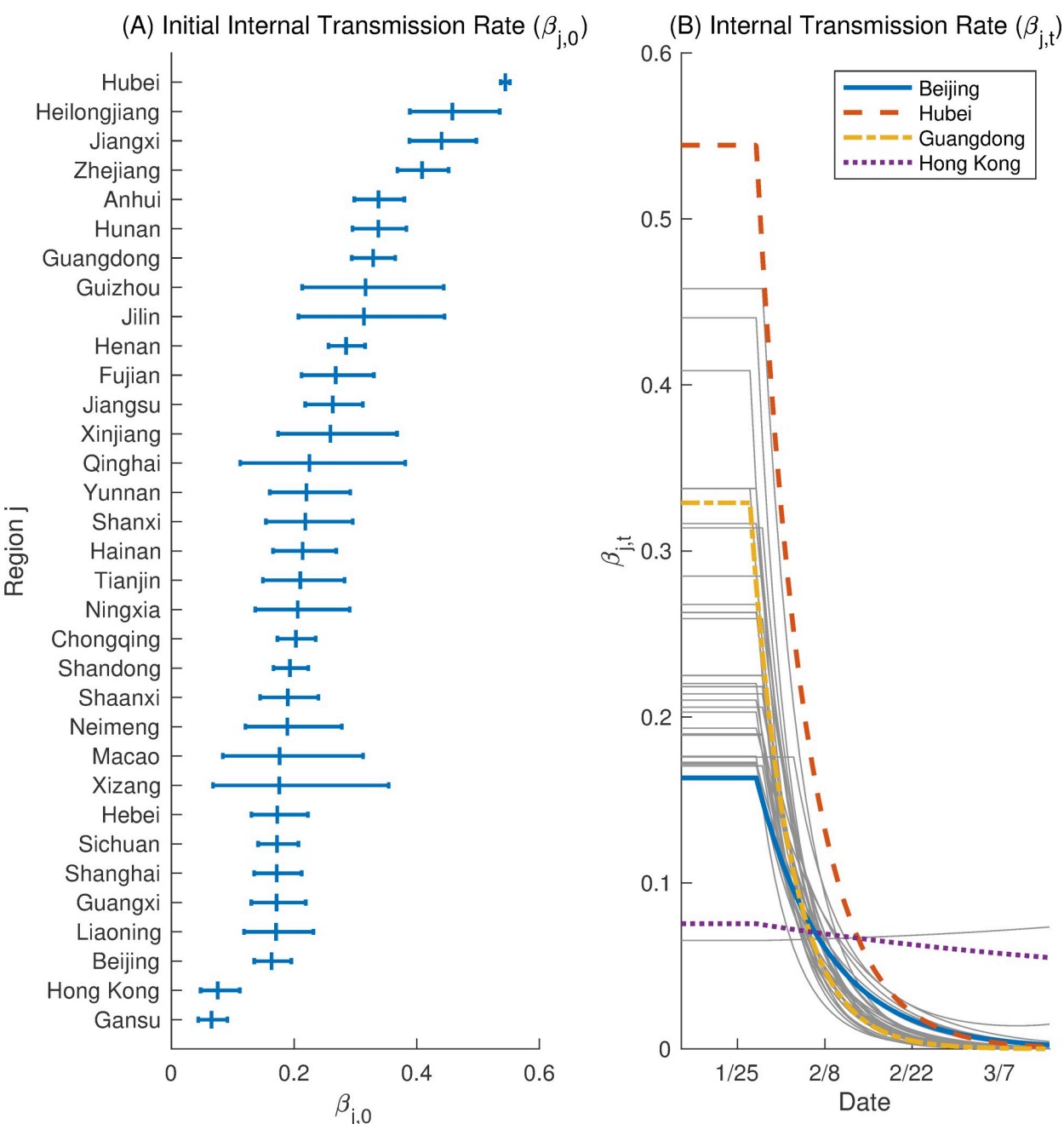

**Fig 2. Internal transmission rate.** Panel (A) shows that the posterior mean of the basic reproduction number for each region with the line segment representing the 95% posterior credible interval. Panel (B) reports the posterior mean of the effective reproduction number ($R_t$) across regions over time.

Internal transmission in Beijing and Guangdong follows a similar pattern with an exponential increase from the beginning of the outbreak, which dominates the external transmission influence. In these regions, there is an initial peak during the Chinese New Year (January 24—February 2, 2020). This finding is empirical evidence that the pandemic had already expanded outside of Wuhan as early as late January 2020. By this time, other major cities can be

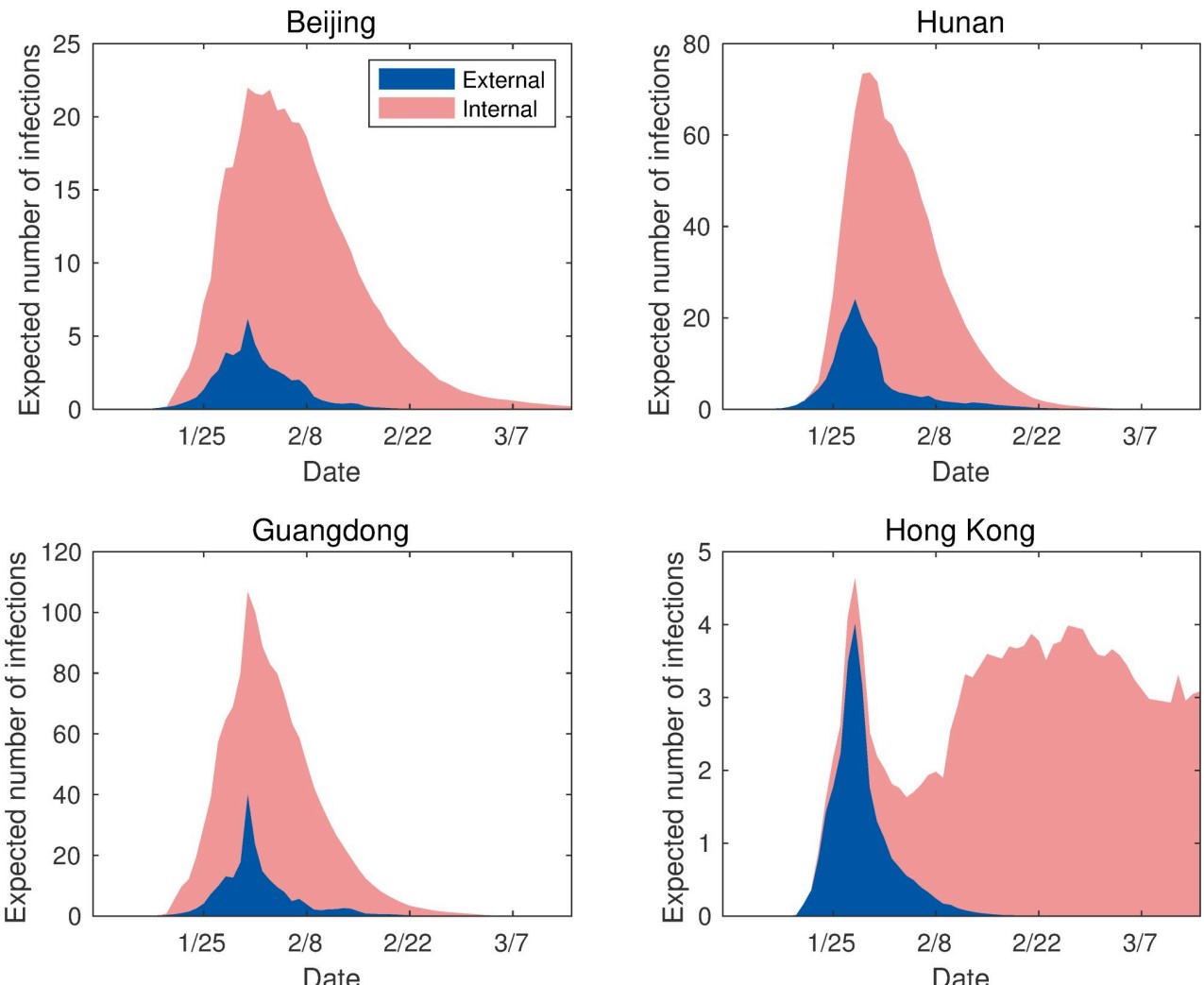

**Fig 3. External vs internal transmission: The number of infected individuals.** The figures show the expected number of infections due to the external infection (the blue area) versus the expected number of infections due to internal infection (the pink area).

considered to have been suffering from localized outbreaks already. On the other hand, the dominating form of transmission in Hunan is external until January 27. Similarly, in Hong Kong, external transmission dominates as the source of infection until February 5. Both internal and external transmission subsequently exhibit an exponential decrease due to unprecedented policy interventions, such as stay-at-home instructions and extended public holidays. The only exception to this observation is the evidence of internal transmission in Hong Kong, which still increased substantially following February 5 until it stabilized on February 15. This can be explained by Hong Kong adopting a different set of policies which were less draconian than the ones adopted in mainland China (see section for details).

## Counterfactual analysis

To shed further light on the effect of mobility, we conduct a counterfactual study to estimate the number of infections if there are no mobility restrictions. Specifically, we assume that the corresponding value can represent the mobility between regions without restriction in 2019.

**Table 1. Number of infections under counterfactual scenarios.**

|  | Beijing | Hunan | Guangdong | Hong Kong | All |
|---|---|---|---|---|---|
| Actual | 446 | 1,018 | 1,361 | 148 | 81,011 |
| Scenario 1 | 638 | 2,261 | 2,375 | 175 | 93,418 |
|  | [567, 712] | [2,139, 2,386] | [2,248, 2,502] | [106, 254] | [89,432, 97,520] |
| Scenario 2 | 1,700 | 8,089 | 8,093 | 417 | 309,287 |
|  | [1,562, 1,844] | [7,777, 8,403] | 7,778, 8,419] | [304, 542] | [298,599, 320,264] |

*Notes*: The row, "actual", reports the actual number of infections. The rows, "Scenario 1" and "Scenario 2", present the average of number of infections over 10,000 simulations with the 90% prediction interval in square brackets. In Scenario 1, we assume that there are no mobility restrictions across regions, while the local policy is placed at the time of the actual data. In Scenario 2, we consider the case where no mobility restrictions across regions are placed and the local policy enacted three days later.

Due to the significant impact of CNY on mobility, we match the dates according to the CNY. For example, the counterfactual mobility on January 23, 2020 (two days before CNY in 2020) is represented by the mobility on February 3, 2019 (two days before CNY in 2019). Using the model and the estimated parameters, we simulate 10,000 paths to compute the average number of infections and the 90% prediction interval. We set the initial state of disease transmission as the realized number of infected, susceptible, dead and recovered individuals from January 11 to January 28 and obtain the evolution of disease transmission until March 15. We choose this period as most regions imposed the local lockdown policy on January 24 or January 25, and the policy takes four days to affect the infection number.

Table 1 reports the number of simulated infections on average and the 90% prediction interval if there are no mobility restrictions across regions. We consider two scenarios, Scenario 1 and 2, both of which pose no mobility restrictions across regions. In Scenario 1, the local policies such as compulsory mask and working from home are activated at the same time as of the actual data. In Scenario 2, those local policies are delayed by three days. From Table 1, if mobility were at the level of 2019, but the local lockdown is in effect, there will be on average a 15% increase in the number of infected individuals across the country (12,407 individuals). Mobility restrictions have a different impact on different regions. For Beijing, the free border would have led to a 43% increase in infections, whereas Hunan—one of the neighbouring region of Hubei—would have seen a 122% increase. For Guangdong and Hong Kong, average increases in infected individuals would be 75% and 18%, respectively. Delaying the local policy has a large effect on infections. A delay of only three days would increase the country's infection number by 282% (228,276 individuals). Regions that already have a large number of infections and/or have close ties with Hubei are significantly affected, such as Hunan (a 695% increase) and Guangdong (a 495% increase).

## Transmission network

The transmission network between the regions of China is observed to evolve on each day of the pandemic [19]. Focuses on the transmission from Wuhan to the rest of China and they conclude that the propagation of COVID-19 in China during the early stage of the outbreak was mostly explained by human mobility originated from Wuhan. However, the authors did not consider the mobility network among the rest of China's geography, and thus, the scope of analysis of the transmission channels is limited. The main advantage of the model developed in this study is that it enables the transmission network to be analyzed on a more granular level. This means that the sources of external transmission and their respective intensities can be identified. Specifically, based on (5), we can obtain the rate of external transmission from

region $k$ to region $j$,

$$A_{jkt} := \frac{\theta_t}{N_t} M^{out}_{k,t-h} P_{kj,t-h} \frac{I_{kt}}{N_k} S_{jt}.$$

Following the literature on network theory [38], we can interpret the square matrix consisting of $A_{jkt}$ for $j, k = 1, \ldots, 33$ as an adjacency matrix of a directed graph with weighted directions $A_{jkt}$ from $k$ to $j$ at time $t$. The sum $\Sigma_{j \neq k} A_{jkt}$ represents the diseases transmissions which originated from region $k$ and moved to the other regions at time $t$.

Fig 4 presents the heatmap of $\Sigma_{j \neq k} A_{jkt}$, thus clarifying the top 10 most influential regions, that is, the regions which were the source of the most transmissions, over time. Hubei stands out as the primary exporter of the infection during the Chinese New Year holidays, though results show that secondary epicenters, such as Beijing, Guangdong, and Shanghai, started being a significant source of transmission from around January 22. The outflux from epicenters, including the primary one, Hubei, gradually diminished following the enactment of policy interventions.

To examine the transmission network more closely, we present a section of the transmission network on January 27, 2020, in Fig 5. All the regions in the network are depicted according to their geographic location. The arrows reflect transmission directions. The lines display transmissions with $A_{kjt}$ greater than two, whereby the line width is proportional to the transmission strength. Fig 5 shows that Hubei is the primary epicenter, particularly for geographically proximate regions (e.g., Henan and Hunan), but also that secondary epicenters such as Beijing, Guangdong, and Shanghai, have already developed on this date. The dynamic migration between the secondary epicenters—which are cultural and economic centers—and the rest of China accelerated the propagation of the disease. Regions such as Shandong and Guangxi are only influenced by secondary epicenters, whereas regions such as Sichuan evidence disease transmission originating from both the primary epicenter and the secondary epicenters.

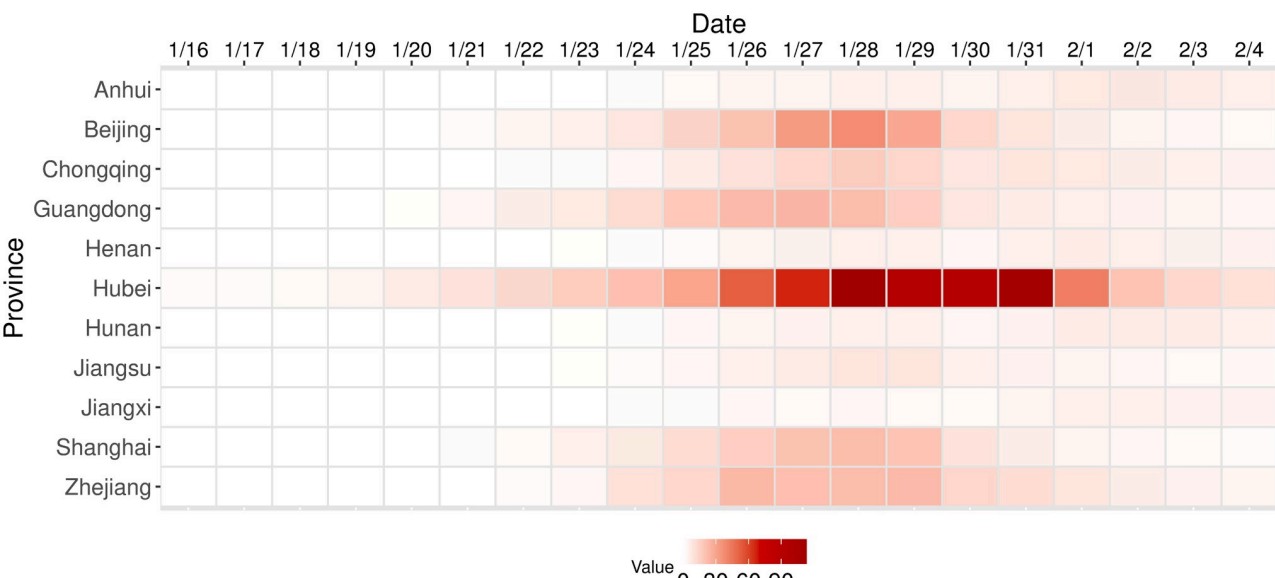

**Fig 4. Origins of transmission: The effective infected outflux from each region over time.** *Notes*: The horizontal axis shows days from January 16 to February 4 and the vertical axis shows ten regions, which are the origins of the ten highest external daily transmission to the other regions. The heatmap reports values of $\Sigma_{j \neq k} A_{jkt}$ for each origin $k$.

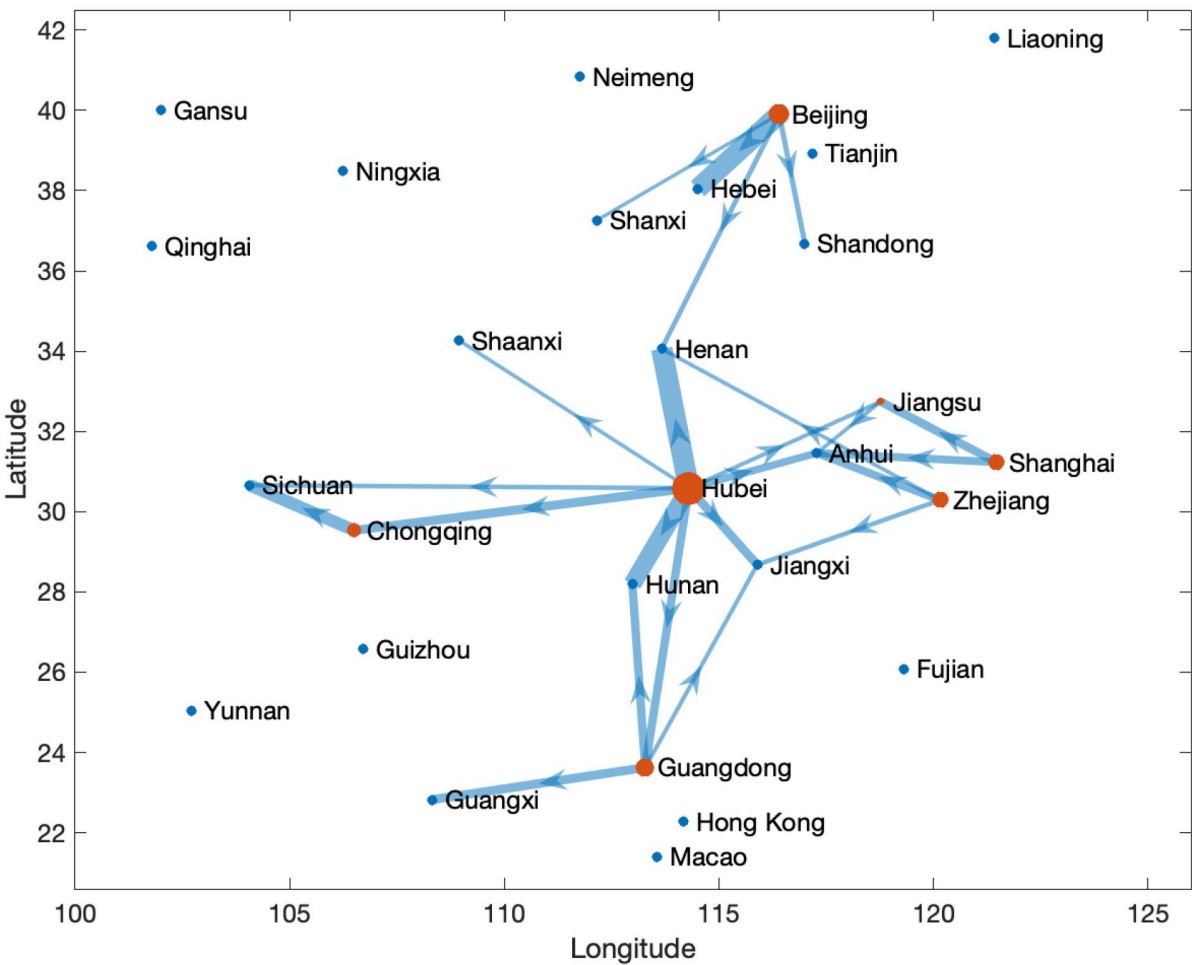

**Fig 5. Transmission network on January 27, 2020.** *Notes*: Regions are located geographically. The arrow indicates the direction of transmission; the lines display external transmissions that are greater than two. The line widths are proportional to the external transmission in the indicated direction, and the size of the nodes is proportional to the total export from a region.

## Discussion and conclusion

### Discussion

In this paper, we analyze the propagation of COVID-19 among 33 regions in China. We develop a spatial model that extends the SIR-type model to estimate the effect of the policy interventions on the disease spread across 33 regions. Our estimation results suggest that secondary epicenters such as Beijing, Guangdong, and Shanghai, developed at a very early stage of the outbreak. Our analysis also shows that mobility restrictions across regions indeed prevented the further spread of the disease. Community transmission was observed to be the primary source of infection, and it declines substantially following local policy interventions.

The epidemiology literature traditionally focuses on deterministic SIR-type models, while our paper extends the standard models to allow for a stochastic mechanism. Also, since the COVID-19 outbreak, a few studies have used the Baidu database to document the disease transmission, but they only focus on the transmission from the primary epicenter, Wuhan, to the rest of China. Our spatial transmission network suggests that other epicenters can quickly become established.

Due to the lack of data availability, our analysis in this paper does not separately evaluate the effects of different policies among regions. If detailed data is available, however, our model does allow such analysis.

## Conclusion

We draw from our empirical results that the Chinese government's responses to the outbreak effectively suppressed the spread of the diseases in China within a couple of months from the start of the epidemic. Our transmission network extracts the cross-region transmission, which, in turn, differentiates the effect of local policies and cross-region mobility restrictions. The spatial analysis allows us to conclude that both local lockdown policies and cross-region mobility restrictions are important for curbing the transmission. To control the disease effectively, the central and local government need to coordinate so that policies can be implemented smoothly and simultaneously.

## Supporting information

**S1 File.**
(ZIP)

## Acknowledgments

The authors like to acknowledge the financial support from the Centre for Development Economics and Sustainability (CDES) at Monash University. Yunyun Wang provided superb research assistance.

## Author Contributions

**Conceptualization:** Tatsushi Oka, Wei Wei, Dan Zhu.

**Data curation:** Tatsushi Oka, Wei Wei, Dan Zhu.

**Formal analysis:** Tatsushi Oka, Wei Wei, Dan Zhu.

**Funding acquisition:** Tatsushi Oka, Wei Wei, Dan Zhu.

**Investigation:** Tatsushi Oka, Wei Wei, Dan Zhu.

**Methodology:** Tatsushi Oka, Wei Wei, Dan Zhu.

**Project administration:** Tatsushi Oka, Wei Wei, Dan Zhu.

**Resources:** Tatsushi Oka, Wei Wei, Dan Zhu.

**Software:** Tatsushi Oka, Wei Wei, Dan Zhu.

**Supervision:** Tatsushi Oka, Wei Wei, Dan Zhu.

**Validation:** Tatsushi Oka, Wei Wei, Dan Zhu.

**Visualization:** Tatsushi Oka, Wei Wei, Dan Zhu.

**Writing – original draft:** Tatsushi Oka, Wei Wei, Dan Zhu.

**Writing – review & editing:** Tatsushi Oka, Wei Wei, Dan Zhu.

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
