## [Decision Letter · Decision Letter 0]

8 Apr 2021

PONE-D-21-04530

The Effect of Human Mobility Restrictions on the COVID-19 Transmission Network in China

PLOS ONE

Dear Dr. Oka,

Thank you for submitting your manuscript to PLOS ONE. After careful consideration, we feel that it has merit but does not fully meet PLOS ONE’s publication criteria as it currently stands. Therefore, we invite you to submit a revised version of the manuscript that addresses the points raised during the review process.

We look forward to receiving your revised manuscript.

Kind regards,

Bing Xue, Ph.D.

Academic Editor

PLOS ONE

Journal Requirements:

In the Methods section please provide a citation for the Baidu mobility data, and please provide further clarification whether this is a publicly accessible database. Furthermore, please specify the date range over which data was extracted.

Thank you for stating in your Funding Statement:

This research is partially funded by the Centre for Development Economics and Sustainability (CDES) at Monash University.

Reviewers' comments:

Reviewer's Responses to Questions

**Comments to the Author**

1. Is the manuscript technically sound, and do the data support the conclusions?

Reviewer #1: Yes

Reviewer #2: Partly

2. Has the statistical analysis been performed appropriately and rigorously? 

Reviewer #1: Yes

Reviewer #2: No

3. Have the authors made all data underlying the findings in their manuscript fully available?

Reviewer #1: Yes

Reviewer #2: Yes

4. Is the manuscript presented in an intelligible fashion and written in standard English?

Reviewer #1: Yes

Reviewer #2: No

5. Review Comments to the Author

Reviewer #1: The paper deals with the mobility restrictions and Covid-19 infections in some regiosn in China from the early days of the epidemic. It is an important topic for future policy making and reviews. The authros have collected reliable data from sources related the epidemic outburst period and made the stochastic analysis with the SIR model. The general conclusion is that the major infections were due to local transmissions, and the mobility restrictions are effective.

The reviewer agrees with the results and conclusions, and the method and data are also relaible and appropirate to make the analysis. The paper is well written in the general design and flow. It is recommended to publish with following revisions:

1. The four special admisnistrative areas in China is offically called Municipalities, which include Beijing, Shanghai, Tianjin, and Chongqing.

2. In Figure 3, the patern in Hong Kong is so different from other regions. Please explain about the pecial reason, possibly the difference on the policy. It is a good point to make comparisons.

3. Are there differences on policies in regions? How about the effectiveness?

4. It is possible to itemize the restrictions like highway, railway, and airlines?

Reviewer #2: General comments:

To my thinking, this paper has not been well organized and written. First, the structure of this paper is incomplete. Second, little new insights into the propagation of COVID-19 among 33 regions in China has been brought to readers.

Therefore, the paper is not acceptable for publication in its present form.

Specific suggestions

1 [The structure of Abstract is not complete] The abstract was not well organized in this paper. A good abstract comprises the following four parts: background, methods, results or findings, conclusions or significance. The components of this paper's abstract is not complete.

2 [Figures ] In order to make the content of a plot clear, the text labels should be added to both the x- and the y- axes (e.g., figure 1-3).

3 [Models] The theoretical basis of the model presented in the paper is not clear. As we know, mathematical models fall into two types: mechanism type and parameter type. Correspondingly, the modeling method can be divided into two categories: analytical method and experimental method. What method does the author used in this article? What model to build? The author's modeling idea and process are not clear enough.

4 [Results and conclusions] Conclusions are different from results. Generally speaking, results come from or are directly based on data analysis. In contrast, conclusions come from discussion and represent the climax of discussion. If the result is regarded as the heart of an academic paper, the discussion can be treated as the nerve center of the paper. Discussions form a bridge between results and conclusions. In this paper, the results are confused with conclusions.

5 [Lack of discussion] The section of Results represents the heart of a paper, and the section of Discussion is the paper’s nerve center. [See: Robert A. Day, Barbara Gastel. How to Write and Publish a Scientific Paper (Sixth Edition). Cambridge University Press, 2003] The analytical process of discussion is not clear. The section of Discussion in a paper is generally involved with 3 or 4 parts: (1) main points, which response to the questions put in introduction; (2) comments on related studies or problems; (3) shortcomings or deficiency in study method or process; (4) conclusions, which can be separated to make the final section. [See: Bjorn Gustavii (2002). How to Write and Illustrate a Scientific Paper. Cambridge: Cambridge University Press]

6 [Conclusions are not well extracted] The conclusions should be made clearer in expressions and meanings. The important conclusions should be given three times in a paper: once in the Abstract, again in the Introduction, and again (in more detail probably) in the Discussion (if it contains a Conclusion paragraph) or Conclusions (if this part is separated as the final section). [See: Robert A. Day, Barbara Gastel. How to Write and Publish a Scientific Paper (Sixth Edition). Cambridge University Press, 2003]

The problem with this article is that there is no significant difference between the section of conclusion part and the section of abstract. As indicated above, a complete abstract comprises four components: background and aim, method, results, and conclusions.

6. PLOS authors have the option to publish the peer review history of their article (what does this mean?). If published, this will include your full peer review and any attached files.

Reviewer #1: **Yes: **Ji Wang

Reviewer #2: No

---

## [Author Response · Author response to Decision Letter 0]

19 May 2021

Please find the attached file, "cover_letter_and_response.pdf", regarding our reponses to the Editor's and two referees' coments.

---

## [Decision Letter · Decision Letter 1]

8 Jun 2021

PONE-D-21-04530R1

The Effect of Human Mobility Restrictions on the COVID-19 Transmission Network in China

PLOS ONE

Dear Dr. Oka,

Thank you for submitting your manuscript to PLOS ONE. After careful consideration, we feel that it has merit but does not fully meet PLOS ONE’s publication criteria as it currently stands. Therefore, we invite you to submit a revised version of the manuscript that addresses the points raised during the review process.

We look forward to receiving your revised manuscript.

Kind regards,

Bing Xue, Ph.D.

Academic Editor

PLOS ONE

Journal Requirements:

Reviewers' comments:

Reviewer's Responses to Questions

**Comments to the Author**

1. If the authors have adequately addressed your comments raised in a previous round of review and you feel that this manuscript is now acceptable for publication, you may indicate that here to bypass the “Comments to the Author” section, enter your conflict of interest statement in the “Confidential to Editor” section, and submit your "Accept" recommendation.

Reviewer #1: All comments have been addressed

Reviewer #2: (No Response)

2. Is the manuscript technically sound, and do the data support the conclusions?

Reviewer #1: Yes

Reviewer #2: (No Response)

3. Has the statistical analysis been performed appropriately and rigorously? 

Reviewer #1: Yes

Reviewer #2: Yes

4. Have the authors made all data underlying the findings in their manuscript fully available?

Reviewer #1: Yes

Reviewer #2: Yes

5. Is the manuscript presented in an intelligible fashion and written in standard English?

Reviewer #1: Yes

Reviewer #2: Yes

6. Review Comments to the Author

Reviewer #1: (No Response)

Reviewer #2: After revision, the quality of this paper has been improved. The author deals with the comments within his ability. I recommend that this paper be published in PLoS ONE with minor amendments.

The results of this paper provide useful information about COVID-19 transmission network in China. However, the discussion part and conclusion part are still not well written.

First, the part of the discussion is incomplete. Please refer to my first round of comments.

Second, the value of this study cannot be seen from the section of conclusions. The conclusion part does not give useful information about COVID-19 transmission network in China.

7. PLOS authors have the option to publish the peer review history of their article (what does this mean?). If published, this will include your full peer review and any attached files.

Reviewer #1: **Yes: **Ji Wang, Ningbo University, wangji@nbu.edu.cn

Reviewer #2: No

---

## [Author Response · Author response to Decision Letter 1]

14 Jun 2021

Please find the attached file, "Cover_Letter_and_Response.pdf", regarding our response to the issue you kindly raised.

---

## [Editor Report · Decision Letter 2]

28 Jun 2021

The Effect of Human Mobility Restrictions on the COVID-19 Transmission Network in China

PONE-D-21-04530R2

Dear Dr. Oka,

We’re pleased to inform you that your manuscript has been judged scientifically suitable for publication and will be formally accepted for publication once it meets all outstanding technical requirements.

Kind regards,

Bing Xue, Ph.D.

Academic Editor

PLOS ONE
---

## [Editor Report · Acceptance letter]

8 Jul 2021

PONE-D-21-04530R2 

The effect of human mobility restrictions on the COVID-19 transmission network in China 

Dear Dr. Oka:

I'm pleased to inform you that your manuscript has been deemed suitable for publication in PLOS ONE. Congratulations! Your manuscript is now with our production department. 

Kind regards, 

on behalf of

Professor Bing Xue 

Academic Editor

PLOS ONE